# Gut over Mind: Exploring the Powerful Gut–Brain Axis

**DOI:** 10.3390/nu17050842

**Published:** 2025-02-28

**Authors:** Stefana-Maria Petrut, Alexandra Maria Bragaru, Alice Elena Munteanu, Adina-Diana Moldovan, Cosmin-Alec Moldovan, Elena Rusu

**Affiliations:** 1Department of Preclinical Sciences, Faculty of Medicine, Titu Maiorescu University, 031593 Bucharest, Romania; stefana.petrut@gmail.com (S.-M.P.); elenarusu98@yahoo.com (E.R.); 2Doctoral School of Medicine, Titu Maiorescu University of Bucharest, 040317 Bucharest, Romania; maria.bragaru@yahoo.com (A.M.B.); rarinca.diana@gmail.com (A.-D.M.); 3Department of Medico-Surgical and Prophylactic Sciences, Faculty of Medicine, Titu Maiorescu University, 031593 Bucharest, Romania; dralicepopescu@yahoo.com; 4Department of Cardiology, “Dr. Carol Davila” Central Military Emergency University Hospital, 010825 Bucharest, Romania; 5MedLife SA, 010719 Bucharest, Romania; 6Department of General Surgery, Witting Clinical Hospital, 010243 Bucharest, Romania

**Keywords:** gastrointestinal tract, probiotics, microbiota, mental health, gut–brain axis

## Abstract

**Background:** The human gastrointestinal tract is home to a wide variety of microorganisms. For some decades now, bacteria known as probiotics have been added to various foods because of their beneficial effects for human health. Evidence indicates that probiotics significantly regulate gut microbiota, which is vital for digestion, metabolism, immune function, and mental health. **Methods:** We conducted a narrative review of available original research published in PubMed for the past ten years focusing on recent advancements that provide a thorough understanding of the relationship between the gastrointestinal system and the brain. **Results:** Recent advances in research have focused on the importance of gut microbiota in influencing mental health. The microbiota–gut–brain axis is a complex, bidirectional communication network linking the central nervous system and the gastrointestinal tract, which highlights how the gut and brain are deeply interconnected and influence each other in ways that affect our overall health, emotions, and behavior. This powerful link is a major area of research as scientists discover more about how gut health can impact mental well-being. **Conclusions:** A comprehensive understanding of microbiota composition and mechanisms involved in these interactions between the gut and the brain could shape future medical and therapeutic approaches. It would balance scientific explanation with clinical relevance, offering insights into how understanding the brain–gut axis can revolutionize our approach to treating mental health and gastrointestinal disorders.

## 1. Introduction

The human gastrointestinal (GI) tract is far more than a digestion hub—it is a dynamic player in mental and emotional well-being. The gastrointestinal tract, as the body’s largest interface with the external environment, has developed a variety of mechanisms to prevent pathogen invasion and efficiently eliminate any pathogens that do enter. Simultaneously, it must maintain tolerance to self-antigens and harmless non-self-entities, including nutrients and the commensal microbiota, to prevent inappropriate immune responses [1].

Probiotics, as defined by the Food and Agriculture Organization/World Health Organization (FAO/WHO) 2002 working group on the evaluation of probiotics in food, are “live microorganisms that, when administered in adequate amounts, confer a health benefit on the host” [2]. These microorganisms contribute to gut microbiota regulation by enhancing the abundance of beneficial bacteria, thereby supporting microbial balance and strengthening immune function [3,4]. Lactic acid bacteria (LAB) are among the most important groups of bacteria, with critical roles in the food, pharmaceutical, and medical industries. The fast-growing characteristics of LAB strains; their metabolic activity, associated with production of many beneficial compounds; and most of all, their GRAS (generally recognized as safe) status recommend them for extensive prophylactic and therapeutic use as probiotics [5,6].

Microbes have always been an essential part of human life. The human gastrointestinal tract is home to the most diverse microbial community in the body, encompassing bacteria, fungi, viruses, and archaea [7,8]. More than 2000 bacterial species have been characterized within the human gut, and it is estimated that the microbial genome of the gut microbiota contains approximately 150 times the number of genes found in the human genome [9,10]. This extensive genetic and metabolic capacity of the gut microbiome is integral to a wide range of physiological processes, including health maintenance, development, aging, and the pathophysiology of various diseases [11].

Although the gut–brain axis and intestinal microbiota have been the focus of intensive research in recent years, interest in this field is not entirely new. Historically, the gastrointestinal system has been recognized as a potential origin of both physical and mental health disturbances, with early medical theories acknowledging its role in overall well-being [12]. The influence of bacteria on our emotions or behavior was observed and led to discovery of the so-called gut–brain axis [13]. This concept refers to the bidirectional communication network that links the central nervous system with the gastrointestinal tract. The routes of this communication are not fully elucidated, but it refers to a complex signaling system that involves direct neural connections, such as the vagus nerve, as well as hormonal, immune, and microbial pathways that allow the brain and gut to influence each other’s function. The brain–gut axis is often studied in relation to conditions such as gastrointestinal disorders, mental health issues, and the impact of the microbiome on overall health [14,15].

Notable biological functions of the gut microbiome include secretion of neuroactive substances, including serotonin, dopamine, and γ-aminobutyric acid (GABA), which influence our behavior, cognition, and emotions. Moreover, gut microbiota is capable of regulation of nutrient harvest from the diet, cholesterol metabolism, modulation of the host’s immune response, and even production of antimicrobial peptides [16].

Accumulating evidence has revealed that the microbiome may serve as a promising therapeutic target, particularly considering emerging findings that probiotic bacteria could offer a viable approach to treating neuropsychiatric symptoms [17]. The mechanisms of action of probiotics are related to their ability to compete with pathogenic microorganisms for adhesion sites, to antagonize these pathogens or to modulate the host’s immune response.

Both in vitro evidence and in vivo clinical data have revealed potential applications of probiotics in the prevention and treatment of various health conditions and diseases. Recent molecular advancements have provided strong indications to support and justify the hypotheses. The potential application of probiotics includes prevention and treatment of various health conditions and diseases, such as gastrointestinal infections, inflammatory bowel disease, lactose intolerance, some allergies, urogenital infections, various cancers, reductions in antibiotic side effects, oral health concerns such as prevention of dental caries, and many other effects that are under investigation. Although the beneficial effects of probiotics have been demonstrated in many studies, more research is needed to evaluate the efficacy and safety of probiotics [18,19,20].

We conducted this narrative review based on original research published in PubMed in the past ten years by selecting only original papers, in English, containing the following keywords: gastrointestinal tract, probiotics, microbiota, mental health, and gut–brain axis. Out paper tries to underscore the emerging scientific understanding of how gut health impacts mental health by exploring the mechanisms underlying this relationship and discussing the potential therapeutic interventions that leverage the gut–brain connection. The aim is to provide readers with an in-depth and accessible overview of the cutting-edge research and clinical implications of the gut–brain axis.

## 2. Mechanisms of Communication

### 2.1. Characteristics of Gut Microbiota

The gut microbiota comprises trillions of microorganisms that live in the gastrointestinal tract, including bacteria, viruses, fungi, and archaea. Over 1000 bacterial species coexist, with dominant phyla being Firmicutes and Bacteroidetes, followed by Actinobacteria, Proteobacteria, and Verrucomicrobia. Microbial diversity differs between individuals and across various regions of the gut. Additionally, the microbiome is a complex concept encompassing not only the microorganisms themselves but their genetic material and the specific environment in which they reside [21,22]. (Table 1).

Gut microbiota has a dynamic composition. Evolving from birth, influenced by delivery mode and breastfeeding, to old age, gut microbiota profile in humans changes with the age of the host and is most stable in adulthood, albeit with reduced diversity. While bacterial species dominate, the composition of microbiota varies significantly between individuals and is shaped by genetics, diet, age, geography, and lifestyle. The key microbial groups and notable species that are commonly found in the human gut are represented by *Lactobacillus* and *Bifidobacterium* species, which are often present in the commercial formulas of many probiotics. Moreover, *Enterococcus* species may be found, and less abundantly, so may *Saccharomyces* and *Candida* species. Furthermore, *Akkermansia muciniphila*, known for mucin degradation or viruses, mainly bacteriophages, and archaea such as *Methanobrevibacter smithii* should be included as well. New species are continually being identified through metagenomic sequencing, and many gut microbes remain uncultured or unclassified [23,24,25].

### 2.2. Vagus Nerve and Enteric Nervous System

The vagus nerve and the enteric nervous system (ENS) are critical components of the gut–brain axis, playing central roles in bidirectional communication between the gastrointestinal tract and the central nervous system (CNS) [26].

The enteric nervous system, also known as the ‘second brain’, is embedded in the walls of the GI tract (esophagus to rectum). It contains approximately 500 million neurons organized into two plexuses: the myenteric (Auerbach’s) plexus, which controls gut motility (peristalsis), and the submucosal (Meissner’s) plexus, linked to regulation of secretion, blood flow, and nutrient absorption [27,28].

The vagus nerve, ‘the gut–brain superhighway’, is the longest cranial nerve and the major communication pathway between the gut and the brain, facilitating the transmission of signals from the gut to the CNS [26].

Bidirectional communication in the gut–brain axis refers to the constant, dynamic exchange of signals between the GI tract (including the gut microbiota and enteric nervous system) and the central nervous system (CNS). Both afferent (gut to brain) and efferent (brain to gut) pathways include the role of the vagus nerve, which is essential, as it is a major communication channel. Furthermore, the ENS plays an important role as a local control system that can operate independently but still communicate with the CNS. The microbiota also plays a part in producing metabolites that influence both the gut and brain. Signals originating in the gut travel to the brain via three main routes: neural pathways, humoral pathways, and immune pathways. The brain sends signals back to the gut to regulate its function via the autonomic nervous system, the hypothalamic–pituitary–adrenal axis, or neurotransmitter release [29,30,31].

### 2.3. Neurotransmitters and Metabolites

The gut–brain axis relies heavily on neurotransmitters (chemical messengers) and metabolites (small molecules produced during metabolism) to facilitate communication among the gut, microbiota, and brain. These molecules influence digestion, immunity, mood, and cognition [32].

Neurotransmitters are produced by host cells and gut microbiota, enabling bidirectional signaling. The main neurotransmitters that may play a role in gut–brain axis communication are serotonin, dopamine, noradrenaline, and gamma-aminobutyric acid (GABA) [33]. These substances are produced not only within the central nervous system but by enteroendocrine cells (EECs), which can synthesize neurotransmitters in response to intestinal peptides, and by the gut microbiota independently. Research has shown that approximately 90% of the body’s serotonin is synthesized by EECs in the gut. As key roles, serotonin regulates gut motility, secretion, and pain perception, as well as modulating mood and cognition via the brain. Serotonin and other neurotransmitters, including dopamine and norepinephrine, are involved in gut–brain communication [34]. Studies have revealed that both dopamine and norepinephrine can be synthesized directly by some species belonging to gut microbiota, such as *Bacillus cereus*, *Bacillus subtilis*, *Staphylococcus aureus*, *Escherichia coli*, or *Proteus vulgaris* [35]. Regarding GABA production, findings indicate that mainly species belonging to the *Bifidobacterium* and *Lactobacillus* genera are directly related to the synthesis of this compound. A study conducted by Bravo et al., 2011, revealed that when administered in mice, *Lactobacillus rhamnosus* led to a reduction in anxiety and depressive-like behavior correlated with an increase in the expression of GABA receptors in the cingulate cortex and a decrease in the expression of GABA receptors in the hippocampus [36,37].

In addition to neurotransmitters, metabolites play a crucial role in gut–brain axis communication. Microbial-derived metabolites such as short-chain fatty acids (SCFAs), tryptophan metabolites, and bile acids act as signaling molecules influencing both gut and brain physiology. SCFAs, including butyrate, acetate, and propionate, modulate neuroinflammation, reinforce the integrity of the blood−brain barrier, and regulate neurotransmitter synthesis. Tryptophan metabolites, such as indole derivatives and kynurenine, impact serotonin production and immune responses, potentially affecting neuropsychiatric conditions. Additionally, secondary bile acids, produced by gut bacteria from primary bile acids, influence neuroinflammation and cognitive function through interactions with receptors such as the farnesoid X receptor (FXR) and Takeda G protein-coupled receptor 5 (TGR5). These metabolites underscore the dynamic role of the gut microbiota in shaping neurophysiology, highlighting their potential therapeutic applications in mental and neurological disorders [38,39].

### 2.4. Immune Pathways

The gut–brain axis employs immune pathways to facilitate bidirectional communication between the gut and the brain that involve immune cells, cytokines, and other molecules in the immune system. The gut and the brain are interconnected through both direct neural connections and immune signaling, which can influence health and disease processes in both systems. Several key immune pathways are involved in the brain–gut axis.

#### 2.4.1. Cytokine Signaling

Immune cells in the gut, such as macrophages and dendritic cells, can produce cytokines that influence the brain’s immune response. These cytokines can travel through the bloodstream or interact with the vagus nerve to impact brain function. For example, pro-inflammatory cytokines such as IL-6 and TNF-α can affect mood and behavior, contributing to conditions such as depression or anxiety [38,39].

#### 2.4.2. Microbial Immune Regulation

The gut microbiome plays a crucial role in modulating immune responses in the gut. The bacteria in the gut can influence the production of immune molecules that affect both gut health and brain function.

In this respect, the immune molecules in question belong to the groups of cytokines, Immunoglobulins, and Pattern Recognition Receptors (PRRs) and Toll-Like Receptors (TLRs).

From the group of cytokines, Interleukin-6 (IL-6), IL-10, IL-17, and Tumor Necrosis Factor-alpha (TNF-α) are especially involved. IL-6 can have both pro- and anti-inflammatory roles, and high levels are linked to neuroinflammation and mood disorders. IL-10 helps maintain gut homeostasis and may protect against neuroinflammation, IL-17 is usually associated with autoimmune diseases and possibly linked to neurodegenerative conditions [37]. TNF-α is a key proinflammatory cytokine involved in gut and brain inflammation; its overproduction is linked to inflammatory bowel disease (IBD), neurodegenerative disorders, and mood disorders by promoting blood−brain barrier disruption and neuroinflammation [38].

Immunoglobulins, especially the secretory IgA (sIgA), protect gut mucosa and modulate immune responses, preventing excessive inflammation that could affect the brain in the long term.

PRRs and TLRs work by activating bacterial lipopolysaccharides (LPS), thus triggering an immune response that can influence neuroinflammation.

Dysbiosis (an imbalance in the gut microbiome) has been linked to altered immune responses that may disrupt brain function and contribute to neurological and psychiatric disorders [40].

#### 2.4.3. Gut–Immune–Brain Communication

Immune cells in the gut can communicate directly with the brain via the vagus nerve, which serves as a major conduit for signaling. For instance, immune-related signals such as inflammatory cytokines or short-chain fatty acids (produced by gut bacteria) can activate the vagus nerve, leading to changes in brain activity and behavior. This communication can affect stress responses, cognition, and mood regulation [41].

#### 2.4.4. Blood−Brain Barrier (BBB) and Immune Modulation

The immune system also plays a role in maintaining the integrity of the blood−brain barrier, which protects the brain from harmful substances. Disruption of this barrier, potentially influenced by immune responses originating in the gut, can allow harmful molecules or immune cells to enter the brain, contributing to neuroinflammation and disorders such as multiple sclerosis or Alzheimer’s disease [42].

In sum, immune pathways within the brain–gut axis highlight the dynamic interaction between the immune system, the gut microbiota, and the brain. These pathways are important for regulating both normal physiological functions (such as digestion and mood) and pathological conditions (such as inflammation, neurological disorders, and mental health issues).

## 3. Impact of Diet and Lifestyle

Diet plays a crucial role in shaping the gut microbiota, which has profound effects on overall health. A diet rich in fiber, prebiotics, and fermented foods can promote a healthy, diverse microbiota, while diets high in fat, sugar, and processed foods may contribute to an imbalance that negatively impacts gut and systemic health. Given the gut microbiota’s significant role in the brain–gut axis and immune function, dietary choices are vital for optimizing both physical and mental well-being [43].

### 3.1. Fiber-Rich Diet

Dietary fibers, found in fruits, vegetables, whole grains, and legumes, act as prebiotics. These fibers are not digested by the human body but serve as food for beneficial gut bacteria. When fermented by gut microbes, fibers produce short-chain fatty acids (SCFAs) such as butyrate, propionate, and acetate, which have anti-inflammatory effects and help maintain gut barrier integrity [44]. A high-fiber diet is generally associated with greater microbiome diversity, which is linked to better overall health. Greater microbial diversity is often considered a hallmark of a healthy microbiota and has been linked to a lower risk of chronic diseases, including obesity, diabetes, and cardiovascular disease [45,46].

### 3.2. Fermented Foods

Fermented foods, such as yogurt, kefir, kimchi, sauerkraut, and kombucha, are rich in live microorganisms, which can add beneficial bacteria (probiotics) directly to the gut. These probiotics can help restore balance to the microbiome, especially in cases of dysbiosis or after antibiotic treatment [47,48]. Regular consumption of fermented foods has been shown to increase gut diversity, improve digestion, and support immune function by promoting the growth of beneficial gut microorganisms [49]. Some studies suggest that dairy products, particularly fermented varieties, can positively influence the gut microbiota by increasing the abundance of beneficial bacteria such as *Lactobacillus* and *Bifidobacterium* species. However, nonfermented dairy products may have mixed effects depending on the individual, as some people may experience negative effects due to lactose intolerance or sensitivity [50].

### 3.3. Probiotics and Psychobiotics

The microbiota appears to be a promising new therapeutic target, especially in the context of the discovery of probiotic bacteria serving as a potential treatment method for neuropsychiatric symptoms.

Probiotics and psychobiotics both refer to live microorganisms that, when consumed in adequate amounts, provide health benefits to the host. While probiotics have long been recognized for their positive effects on gut health, psychobiotics are a more recent concept that extends the potential benefits of probiotics to mental health and brain function, particularly in the context of the brain–gut axis [51]. Dietary supplementation is gaining more popularity, and certain vitamins and minerals have been revealed to have important benefits in the equilibrium of the gut–brain axis. A supportive key element of the gut–brain axis and an adjuvant of probiotics and psychobiotics is magnesium, and especially magnesium orotate, because its benefits have been recently suggested not only in both enteric and microbiome dysfunctions but in some psychiatric disorders [52].

Probiotics are commonly found in fermented foods (such as yogurt, kefir, kimchi, and sauerkraut) and dietary supplements. Probiotics work by restoring or maintaining a healthy balance of gut bacteria, especially when the gut microbiome is disrupted (such as after antibiotic use, illness, or poor diet). While the mechanisms of action of probiotics are actively being investigated, a significant body of knowledge exists regarding the various effects they exert at the host organism level, such as immune system modulation, digestive health, gut microbiota balance, etc. As previously mentioned, the term ‘psychobiotics’ was coined to describe probiotics that can produce positive effects on mental health, particularly by reducing symptoms of anxiety, depression, and stress. Specific strains, such as *Lactobacillus rhamnosus* and *Lactobacillus helveticus*, have been shown to improve symptoms of anxiety and depression in animal models and some human studies. Another important genus of bacteria, with strains such as *Bifidobacterium longum* and *Bifidobacterium infantis*, has been found to improve both gut health and mental well-being by reducing symptoms of depression and enhancing mood by influencing gut-derived serotonin production. Moreover, studies have mentioned a probiotic yeast, *Saccharomyces boulardii*, that has been shown to help with gastrointestinal disorders and may have an impact on mood regulation and stress resilience [53,54,55].

### 3.4. Lifestyle Factors

Lifestyle factors play a significant role in shaping the gut microbiota and influencing the brain–gut axis. Several lifestyle habits and behaviors can impact the composition, diversity, and function of the gut microbiome, which in turn can influence gut–brain communication. Emerging research suggests that fasting or intermittent fasting can lead to beneficial changes in the microbiota, such as increased levels of gut bacteria associated with improved metabolism and reduced inflammation. Fasting periods may allow for ‘gut reset’, reducing dysbiosis and promoting the growth of beneficial species [56,57].

Regular physical activity has been shown to positively influence the gut microbiota by increasing microbial diversity and promoting the growth of beneficial bacteria. Exercise also promotes gut motility, which helps maintain healthy digestion and reduces the risk of constipation.

Poor sleep or irregular sleep patterns from various causes, including low oxygen levels during sleep or sleep-disordered breathing, can disrupt the gut microbiota, leading to a decrease in microbial diversity and potentially favoring the growth of harmful bacteria. Research suggests that sleep disturbances, such as insomnia or shift work, can negatively impact gut health. Thus, inflammation associated with hypoxia could alter gut microbiota composition, which in turn might affect brain function and mental health, as well as contribute to gastrointestinal symptoms such as bloating, pain, and altered motility [58].

Chronic stress, anxiety, and depression can have profound effects on the gut microbiome, leading to an imbalance that may exacerbate digestive issues and increase inflammation. Moreover, stress or anxiety during surgery or of patients during the perioperative period can affect the immune system and determine changes in the gut microbiota composition, potentially reducing beneficial bacteria and promoting the growth of harmful microbes. Additionally, if the surgical procedure leads to complications (for example, bile duct injuries), there may be an impact on overall gut function and digestion. This research may not seem to directly touch on the brain–gut axis, but there may be some indirect connections, as these factors can activate the autonomic nervous system, potentially influencing gastrointestinal functions [59,60].

As a result, maintaining a balanced microbiome is crucial for optimal physical and mental health, as disruptions in the microbiome can contribute to various health issues. By adopting healthy habits such as a balanced diet, regular physical activity, adequate sleep, and stress management, individuals can support a healthy brain–gut axis and overall well-being.

## 4. Clinical Implications

The brain–gut axis has profound clinical implications across a broad range of conditions, from gastrointestinal disorders to mental health issues and neurodegenerative diseases. As our understanding of this complex system grows, it opens new avenues for treatment, including dietary interventions, probiotics, prebiotics, and stress management strategies. By targeting both gut health and brain function, we may be able to offer more holistic, effective treatments for a variety of conditions that involve both the gut and the brain. The integration of gut microbiome analysis into clinical practice could also pave the way for more personalized approaches to healthcare, optimizing outcomes for patients based on their individual microbiome and brain–gut interactions.

### 4.1. Gastrointestinal Disorders

Some authors described a certain level of clinical interconnectivity between signs and symptoms of gut microbiota disorder and dysplastic colonic polyps, a situation that was alleviated partially after repeated hot polypectomies [61].

Irritable bowel syndrome (IBS) is a functional GI disorder characterized by symptoms such as abdominal pain, bloating, and irregular bowel movements. It has a strong link to mental health conditions such as anxiety and depression. Dysfunction in the brain–gut axis may contribute to the development and persistence of IBS symptoms [60]. Mental health treatments such as cognitive behavioral therapy (CBT), mindfulness, and antidepressants have been shown to improve IBS symptoms, suggesting a central role for the brain–gut connection [62].

Inflammatory bowel disease (IBD) includes conditions such as Crohn’s disease and ulcerative colitis that involve chronic inflammation in the GI tract. Dysregulation of the brain–gut axis may influence the immune system, contributing to the exacerbation of IBD. The relationship among the gut microbiome, stress, and immune response in IBD is an area of increasing research. Psychosocial stress management and therapies such as mindfulness and biofeedback may help reduce flare-ups [63].

Other studies have even suggested that a severe impact on the bile fluid inflow at the small intestine site, such as those that may derive from a severe iatrogenic bile duct injury that has, as a means of repair, high choledochal–jejunal anastomosis, may also trigger an important dysfunction in intestinal microbiota, a pathology that can be irreversible [60].

Neuroendocrine neoplasms may also trigger a deep dysfunction in the gut microbiota, a clinical situation observed in some patients enrolled in long-term observation lists after liver resection surgery for their hepatic metastases [64].

Acute pancreatitis and its overwhelming consequences on the entire intestinal ecosystem may also trigger a deep and long-lasting effect on gut microbiota, perhaps due to a severe enzymatic disbalance that follows even after the pancreatic parenchima is restored, to some extent, to its original status and function [65].

### 4.2. Mental Health Disorders

There is growing evidence that disturbances in the gut microbiota and brain–gut communication may contribute to the development and progression of anxiety and depression. The gut microbiota can influence brain chemistry, including the production of neurotransmitters like serotonin (which is primarily produced in the gut). Psychobiotics, or probiotics with effects on the brain, are being investigated as potential adjunct therapies for anxiety and depression, offering a novel approach to treating these conditions. Additionally, gut inflammation may play a role in mood disorders, and targeting inflammation through dietary interventions, probiotics, or anti-inflammatory drugs could become an important therapeutic strategy [66].

Additionally, there are studies that suggest that individuals with autism spectrum disorder (ASD) often have gastrointestinal symptoms and altered gut microbiota. This has led to growing interest in the potential role of the brain–gut axis in autism. Interventions targeting the microbiota, such as probiotics or dietary changes, have shown some promise in improving both gut health and behavior in individuals with ASD. Research into the gut–brain connection in autism is still in its early stages, but it highlights a potential new avenue for treatment [67].

### 4.3. Neurodegenerative Disorders

Emerging research suggests that the gut–brain axis may play a significant role in the pathogenesis of neurodegenerative diseases like Alzheimer’s and Parkinson’s. Both types of diseases have been associated with gut microbiota dysbiosis, and early gastrointestinal symptoms often precede neurological manifestations. It is hypothesized that alterations in gut bacteria may influence brain function through immune signaling, inflammation, and the production of neurotoxic metabolites. Targeting the gut microbiota with probiotics, prebiotics, or dietary modifications could potentially serve as a therapeutic strategy to modulate neurodegenerative disease progression [68,69].

### 4.4. Therapeutic Approaches

Therapeutic approaches targeting the brain–gut axis are an emerging and promising area of research, with implications for treating a wide range of both gastrointestinal and mental health disorders. Various therapeutic strategies are being explored to modulate this axis and improve health outcomes, from dietary changes, probiotics administration, and psychological interventions to pharmacological treatments, exercise, or emerging therapies such as fecal microbiota transplantation.

As research continues to expand our understanding of the brain–gut axis, we can expect more personalized and effective treatments that target the root causes of a wide range of conditions, improving quality of life for many individuals. Fecal microbiota transplantation (FMT) involves transplanting stool from a healthy donor into the gut of a patient with a disrupted microbiome. It has shown significant success in treating *Clostridium difficile* infections and is being explored for other conditions such as IBS, IBD, and even neurological disorders such as Parkinson’s disease. FMT could potentially restore a healthy microbiome, improving both gut and mental health [70,71].

## 5. Emerging Research

Emerging research on the brain–gut axis is rapidly expanding, uncovering new and exciting possibilities for therapeutic interventions that target both gastrointestinal and mental health disorders. The bidirectional communication between the gut and the brain has gained considerable attention in recent years, and scientists are increasingly recognizing the critical role the microbiome plays in this relationship.

### 5.1. Personalized Medicine

As research into the brain–gut axis deepens, there is potential for personalized treatment strategies based on an individual’s microbiome profile. By analyzing an individual’s gut microbiome, clinicians may be able to tailor interventions such as probiotic supplementation, dietary modifications, or microbiome-based therapies to optimize health outcomes, particularly for mental health and gastrointestinal disorders.

The use of specific probiotic strains for treating mental health conditions represents a promising area of clinical research. Psychobiotics have the potential to be tailored to an individual’s gut microbiome composition and genetic profile, enabling a more precise and personalized therapeutic approach.

The relationship between the gut microbiome and the efficacy of pharmaceutical treatments is a growing area of research. Studies are exploring how the microbiome affects the metabolism of drugs, including antidepressants, antipsychotics, and anti-inflammatory medications. This could lead to more effective, individualized treatments that take both genetics and microbiome into account.

Although research into the brain–gut axis is an exciting and rapidly growing field, it is also fraught with several challenges that complicate the discovery of effective therapeutic strategies. The complexity of the brain–gut communication system, the diversity of individual microbiomes, and the need for precise and reproducible methodologies are some of the key hurdles that researchers must overcome [72].

The gut microbiota is highly individualistic, with substantial variation between people in terms of microbial composition and diversity. Factors such as genetics, diet, environment, medications, and lifestyle contribute to this variability. This makes it challenging to generalize findings from animal models or small human cohorts to broader populations. Understanding the relationship between microbiota diversity and health outcomes is complicated by this variability, making it difficult to identify universally applicable therapeutic approaches [73].

### 5.2. Identifying the Correct Physiopathological Processes

Another central challenge in microbiome research is distinguishing between correlation and causation. While numerous studies have linked specific microbiota patterns with conditions such as depression, anxiety, and IBS, it remains unclear whether these microbial changes are a cause of the disorder or a consequence of it. Longitudinal studies and more sophisticated experimental models are needed to establish direct causative links between the microbiome and disease processes. There is no universally agreed-upon standard for analyzing microbiome data.

Animal models are essential for understanding the underlying mechanisms of disease and testing therapeutic approaches. However, the complexity of the brain–gut axis means that most animal models fail to fully replicate human gastrointestinal and mental health conditions. While germ-free mice and microbiome-modulated animals are valuable research tools, they do not completely capture the wide diversity of human microbiomes or the intricate nature of human mental health disorders. Even when animal models do successfully mimic certain aspects of disease, translating findings from animals to humans is often difficult. For instance, results from rodent studies on probiotics or prebiotics might not always translate into positive outcomes in human clinical trials. This is because human physiology, gut microbiota, and the brain–gut axis can be quite different from those of animals, especially in terms of the gut microbiome’s diversity. While early-stage clinical trials have shown promise for gut-based interventions (such as probiotics and fecal microbiota transplantation), many studies yield inconsistent results. Variability in clinical trial design, patient selection, microbiome profiling methods, and even the types of probiotics used may account for these inconsistencies [74,75].

The challenge lies in identifying which treatments work for which patients, as the microbiome’s role in disease may vary greatly from one person to another. This variability makes it difficult to assess the specific impact of microbiome-targeted treatments on distinct subgroups of patients. More refined clinical trial designs, including personalized or stratified approaches, are needed to determine how different microbiome-based interventions may work in specific groups of people.

## 6. Conclusions

Emerging evidence underscores the critical role of the gut microbiome in modulating brain function, influencing mood, cognition, and behavior, as well as contributing to the pathophysiology of various mental health and gastrointestinal disorders. This intricate relationship highlights the potential of microbiome-targeted interventions, including probiotics, prebiotics, diet, and lifestyle modifications, in improving both gastrointestinal and mental health outcomes.

Despite significant advancements, several challenges remain in understanding the precise mechanisms underlying brain–gut communication, including the variability of individual microbiomes, the complexity of microbial metabolites, and the difficulty in translating animal model findings to human clinical settings. Future research should focus on elucidating the molecular pathways that mediate brain–gut interactions, identifying biomarkers for personalized treatments, and exploring the therapeutic potential of microbiome-based therapies in both gastrointestinal and neuropsychiatric disorders.

In conclusion, the brain–gut axis offers a promising avenue for therapeutic innovation, and further exploration may lead to more targeted, effective interventions for a wide range of conditions. However, continued interdisciplinary collaboration and rigorous clinical trials are essential to unlock the full potential of these emerging therapies and ensure their safety and efficacy in diverse patient populations.

## Figures and Tables

**Table 1 nutrients-17-00842-t001:** Key mechanisms of the gut–brain axis and their implications.

Mechanism	Description	Key Implications
Neural Pathways	The vagus nerve and enteric nervous system (ENS) provide direct communication between the gut and brain.	Regulates digestion, influences mood and behavior, and mediates stress responses.
Microbial Metabolites	Gut microbiota produce short-chain fatty acids (SCFAs), neurotransmitters, and other bioactive compounds.	SCFAs modulate inflammation and blood−brain barrier integrity; neurotransmitters influence mood and cognition.
Neurotransmitters	Serotonin, dopamine, gamma-aminobutyric acid (GABA), and norepinephrine are produced by gut microbes and enteroendocrine cells.	Affects mood regulation, cognitive function, and stress resilience.
Immune System Modulation	Gut microbiota interact with immune cells, influencing cytokine production and inflammatory responses.	Chronic gut inflammation is linked to neurodegenerative and psychiatric disorders.
Endocrine Signaling	The hypothalamic–pituitary–adrenal (HPA) axis is influenced by gut microbiota through cortisol and other stress hormones.	Dysregulation is associated with anxiety, depression, and irritable bowel syndrome (IBS).
Diet and Lifestyle Factors	Dietary fiber, fermented foods, probiotics, and exercise impact gut microbiota composition and function.	These modifiable factors can improve gut and mental health, potentially preventing neuropsychiatric conditions.

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
