# Peer review of "Gut over Mind: Exploring the Powerful Gut–Brain Axis"

_nutrients, 2025, doi:10.3390/nu17050842_

Round 1
Reviewer 1 Report
Comments and Suggestions for Authors
The manuscript of the review entitled "Gut over Mind: Exploring the Powerful Gut-Brain Axis", by SM Petrut 1 and co-workers, is an interesting narrative review, based on data obtained from PubMed, on the relationships of gut and brain, a frequent and recurring theme (with variants) in Nutrients because the Special Issue on the bidirectional gut microbiota brain axis relationships.
The methodology of the review is well done, the manuscript is clear and well written. However, I would like to encourage the authors to introduce some illustration-outline that facilitates the reading of the text.
Author Response
Comments 1: However, I would like to encourage the authors to introduce some illustration-outline that facilitates the reading of the text.
Response 1: Thank you for taking the time to review our manuscript and provide this feedback. We propose a table to better synthetize the key aspects mechanisms of the gut-brain axis. It was inserted in LINE 127 (please see the attached revised form of the manuscript). Thank you for supporting our manuscript.

Reviewer 2 Report
Comments and Suggestions for Authors
Re: Manuscript ID: nutrients-3478618
This is a short review dealing with the gut-brain axis. This topic is not new. The article has a very general feature. The focus is not clear. The arguments are not well organized in the different paragraphs. Changes and comments to improve the paper are suggested.
Microbiota and microbiome indicate two different concepts. Authors must use them appropriately.
The focus of the paper is not clear. Mental health or other pathologies? Probiotics or other therapeutical strategies?
Line 133. Abbreviation of gastrointestinal tract (GT) must be anticipated. Check other abbreviations.
Line 161. Replace “enteroendocrine (EEC) cells” with “enteroendocrine cells (EECs)”.
Section 2.3. Metabolites are mentioned but not discussed.
Line 188. Which immune molecules?
Lines 359 and 369. The concept of emerging research is anticipated with respect to paragraph 5.
Line 390. Personalized medicine is the only emerging research?
A figure/table would be appreciated by the reader.
Author Response
Thank you for taking the time to review our manuscript and provide such valuable insights.
Comment 1: This is a short review dealing with the gut-brain axis. This topic is not new. The article has a very general feature. The focus is not clear.
Response 1: This article falls under the category of “descriptive review”, hence the general addressing. Since this is a literature descriptive review, the topic is, indeed, not new, but it provides an up-to-date view of the current research in the gut-brain axis.
Comment 2: The arguments are not well organized in the different paragraphs.
Response 2: We managed to redo the sections and subsections and not the article has 6 sections and subsections to group ideas accordingly.
Comment 3: Line 133. Abbreviation of gastrointestinal tract (GT) must be anticipated. Check other abbreviations.
Response 3: Thank you for pointing that. GI has been defined in LINE 41, in the INTRODUCTION section. All other abbreviations have now been checked and are properly explained on first usage.
Comment 4: Line 161. Replace “enteroendocrine (EEC) cells” with “enteroendocrine cells (EECs)”.
Response 4: Thank you for highlighting the mistake. It has been corrected.
Comment 5: Section 2.3. Metabolites are mentioned but not discussed.
Response 5: Indeed, the section about metabolites is missing entirely. We updated this section to reflect the role of metabolites in the brain-gut axis. LINES 177-189 have been added.
Comment 6: Line 188. Which immune molecules?
Response 6: Indeed, the immune molecules have not been properly discussed. A new section, about the impact of immune molecules, was added, as it was missing. LINES 190-212 have been added.
Comment 7: Lines 359 and 369. The concept of emerging research is anticipated with respect to paragraph 5.
Response 7: The concept of emerging research is anticipated with respect to paragraph 5.”. The emerging research that we refer here are the ones cited in citation index 69 and 70, as it deals with neurodegenerative disorders and new therapeutic approaches (71 and 72). The overall emerging research is described in detail in Section 5.
Comment 8: Line 390. Personalized medicine is the only emerging research?
Response 8: Indeed, it looks like it is the only one, but it is not. We did not format this section properly and thus it only results that personalized medicine is the only emerging research. We labeled properly each emerging research in part, as subchapters to SECTION 5, to better reflect the information.
Comment 9: A figure/table would be appreciated by the reader.
Response 9: Indeed, the manuscript lacks a form of compact-delivered information. We drafted a table to better highlight the key aspects of mechanisms involved in brain-gut axis interaction. The table (table 1) was inserted in LINE 113.

Reviewer 3 Report
Comments and Suggestions for Authors
Gut over Mind: Exploring the Powerful Gut-Brain Axis
L22-24: revise. Too long sentence
L25: provide more details about the methods used in this review.
L42: be consistent with the abbreviations, gastrointestinal (GI).
L48-51: revise its not clear. Probiotics, defined according to Food and Agriculture Organization/World Health 48 Organization (FAO/WHO) 2002 working group on the evaluation of probiotics in food, as”live microorganisms that, when administered in adequate amounts, confer a health benefit on the host”
L66-68: revise, not clear
L112-114: revise. Diversity varies between individuals and gut regions. On the other hand, microbiome is a complex concept which refer to microorganisms, their genomes, and the habitat they reside in.
L121: revise. which are often mentioned in probiotics.
L122: revise
L181-182: revise. Several key immune pathways are involved in the brain-gut axis: 181 Cytokine Signaling: Immune cells in the gut
L186: Microbial Immune Regulation: put it at beginning of the sentence. This applies for all subtitles
Define Psychobiotics
L396-399: avoid repetition of sentences. This applies here and in other parts of the review.
Comments on the Quality of English Language
The review has to be checked for writing techniques. There are some repetitions of sentences and ideas. Be consistent with abbreviations.
Author Response
Thank you for taking the time to review our manuscript and provide these valuable observations.
Comment 1: L22-24: revise. Too long sentence
Response 1: We have rephrased the sentence to provide more clarity in a concise text the methods used in this review.
Comment 2: L25: provide more details about the methods used in this review.
Response 2: the constraints of the ABSTRACT section, 250 words, does not allow for a more detailed explanation regarding the methods used to draft this review. However, we provided several more details in LINE 110. If, however, you consider that a further refinement of the ABSTRACT is in order, we will happly provide one.
Comment 3: L42: be consistent with the abbreviations, gastrointestinal (GI).
Response 3: Thank you for pointing that our. We changed all instances to GI.
Comment 4: L48-51: revise its not clear. Probiotics, defined according to Food and Agriculture Organization/World Health 48 Organization (FAO/WHO) 2002 working group on the evaluation of probiotics in food, as”live microorganisms that, when administered in adequate amounts, confer a health benefit on the host”
Response 4: Than you. We amended the text for better clarity.
Comment 5: L66-68: revise, not clear
Response 5: We amended the text to provide better clarity to the reader.
Comment 6: L121: revise. which are often mentioned in probiotics.
Response 6: We amended the text so that it would provide better clarity.
Comment 7: L122: revise
Response 7: We amended the text so that it would provide better clarity.
Comment 8: L181-182: revise. Several key immune pathways are involved in the brain-gut axis: 181 Cytokine Signaling: Immune cells in the gut
Response 8: We have revised the section.
Comment 9: L186: Microbial Immune Regulation: put it at beginning of the sentence. This applies for all subtitles
Response 9: Thank you. We have revised.
Comment 10: Define Psychobiotics
Response 10: Psychobiotics have been defined in LINE 291. However, if the reviewer believes that a more detailed definition is necessary, we will provide one.
Comment 11: L396-399: avoid repetition of sentences. This applies here and in other parts of the review.
Response 11: We amended the text to avoid repetition and provide clarity.

Round 2
Reviewer 2 Report
Comments and Suggestions for Authors
Re: Manuscript ID: nutrients-3478618
The manuscript has been carefully revised, and it is now recommended for publication.
Reviewer 3 Report
Comments and Suggestions for Authors
Gut over Mind: Exploring the Powerful Gut-Brain Axis
Thank you for providing a revised version.
No further comments